# *FERMT1* Is a Prognostic Marker Involved in Immune Infiltration of Pancreatic Adenocarcinoma Correlating with m^6^A Modification and Necroptosis

**DOI:** 10.3390/genes14030734

**Published:** 2023-03-16

**Authors:** Qian Wu, Jin Li, Pei Wang, Qihang Peng, Zhongcui Kang, Yiting Deng, Jiayi Li, Dehong Yan, Feng Ge, Ying Chen

**Affiliations:** 1College of Life Science, Yangtze University, Jingzhou 434025, China; 2Institute of Biomedicine and Biotechnology, Shenzhen Institutes of Advanced Technology, Chinese Academy of Sciences, Shenzhen 518055, China; 3State Key Laboratory of Freshwater Ecology and Biotechnology, Institute of Hydrobiology, Chinese Academy of Sciences, Wuhan 430072, China

**Keywords:** *FERMT1*, pancreatic adenocarcinoma, prognosis, immune infiltration, m^6^A modification, necroptosis

## Abstract

As an important member of the kindlin family, fermitin family member 1 (FERMT1) can interact with integrin and its aberrant expression involves multiple tumors. However, there are few systematic studies on *FERMT1* in pancreatic carcinoma (PAAD). We used several public databases to analyze the expression level and clinicopathological characteristics of *FERMT1* in PAAD. Meanwhile, the correlation between *FERMT1* expression and diagnostic and prognostic value, methylation, potential biological function, immune infiltration, and sensitivity to chemotherapy drugs in PAAD patients were investigated. *FERMT1* was significantly up-regulated in PAAD and correlated with T stage, and histologic grade. High *FERMT1* expression was closely connected with poor prognosis and can be used to diagnose PAAD. Moreover, the methylation of six CpG sites of *FERMT1* was linked to prognosis, and *FERMT1* expression was significantly related to N6-methyladenosine (m^6^A) modification. Functional enrichment analysis revealed that *FERMT1* co-expression genes participated in diverse biological functions including necroptosis. In addition, the expression of *FERMT1* was associated with immune cell infiltration and the expression of immune checkpoint molecules. Finally, *FERMT1* overexpression may be sensitive to chemotherapy drugs such as Palbociclib, AM-5992 and TAE-226. *FERMT1* can serve as a diagnostic and prognostic marker of PAAD, which is connected with immune cell infiltration and the modulation of m^6^A and necroptosis.

## 1. Introduction

Pancreatic adenocarcinoma (PAAD) is a cancerous and invasive tumor of the digestive system, which causes almost the same number of deaths as cases and ranks as the fourth most common cause of cancer-related fatality [1,2]. With inapparent early symptoms, the majority of patients with PAAD are detected at the advanced or metastatic stage, and the overall prognosis is poor [3,4]. At present, surgical resection is the only feasible treatment, but the recurrence rate after operation is very high, and the 5-year survival rate is approximately 8% [5]. The carbohydrate antigen 19-9 (CA19-9) is the main biomarker currently used for the diagnosis of PAAD, but its low specificity and low sensitivity cannot meet clinical needs [6]. Therefore, there is an urgent need to develop novel promising early biomarkers to strengthen the predictability of PAAD.

The Kindlins are a family of focal adhesion proteins with conserved FERM (4.1-ezrin-ridixin-moesin) domains, and perform many important biological functions including cell adhesion, migration, assembly of the extracellular matrix, proliferation and differentiation [7]. As a member of Kindlins, fermitin family member 1 (FERMT1, also known as Kindlin-1) is related to Kindler syndrome (KS), a genetic disorder that mainly affects the skin and intestine [8]. Moreover, abnormal *FERMT1* expression has been described in several cancers, including colon cancer [9], gastric cancer [10], oral squamous cell carcinoma [11] and nasopharyngeal carcinoma [12]. *FERMT1* is considered to be involved in tumor proliferation, apoptosis, metastasis, and tumor angiogenesis [7]. Recently, one study showed that the knockdown of *FERMT1* expression restrained the migration and invasion in PAAD cells [13]. As an integrin-binding protein, FERMT1 can interact with integrin-linked kinase (ILK) [14], which plays an essential role in mediating the necroptosis of kidney collecting duct epithelial cells, implying that FERMT1 may be associated with necroptosis [15]. In pancreatic cancer, FERMT1 has been identified as a methylation-driving gene to construct a nomogram model to predict overall survival [16]. However, no research has been conducted on the correlation between *FERMT1* and the prognosis of PAAD, and the role of *FERMT1* in PAAD is still ambiguous.

In the present study, we used a variety of databases to analyze the expression of *FERMT1* in pan-caner and evaluated its significance in diagnosing and predicting the prognosis of PAAD. Moreover, we demonstrated the relationship between *FERMT1* expression and DNA methylation, N6-methyladenosine (m^6^A) modification, immune infiltration and the sensitivity of PAAD patients to chemotherapy drugs. Our results indicated that *FERMT1* may be applied as a prospective prognostic biomarker and a novel immunotherapy candidate for patients with PAAD.

## 2. Materials and Methods

### 2.1. Expression Analysis of FERMT1

The expression of *FERMT1* in pan-cancer was evaluated by TIMER2.0 [17] (http://timer.cistrome.org/, accessed on 9 September 2022). Subsequently, *FERMT1* expression in PAAD patients was validated using the UCSC XENA database [18] (https://xenabrowser.net/datapages/, accessed on 10 September 2022) with data from The Cancer Genome Atlas (TCGA) and the Genotype-Tissue Expression (GTEx, https://www.gtexportal.org/home/index.html, accessed on 9 September 2022) project, including 179 PAAD samples with clinical information and 173 normal tissue samples. The data were processed by Toil [19] and finally converted to log2 (TPM + 1) for further analysis. The GSE28735 and GSE62452 datasets, retrieved from the GEO database (https://www.ncbi.nlm.nih.gov/geo, accessed on 10 September 2022), were used to verify the *FERMT1* expression level in PAAD, with fold change > 2, *p* < 0.05 selected as differential expression. Moreover, the immunohistochemistry images of FERMT1 were obtained from the Human Protein Atlas [20] (HPA, https://www.proteinatlas.org/, accessed on 10 September 2022) database to demonstrate the protein expression levels in PAAD tissue and adjacent tissues. We used the R software “limma” package to perform variance analysis on the GEO data. Differential genes were screened by adjusted *p* < 0.05 and |log fold change (logFC)| ≥ 1, and 1 ≤ |logFC| < 2 represented “normal DEGs”, |logFC| ≥ 2 was defined as “super Anno”.

### 2.2. Prognosis and Diagnostic Analysis of FERMT1 in PAAD

The clinicopathological characteristics and gene expression data of the PAAD patients were downloaded from TCGA and divided into low- and high-expression groups according to the median expression value of *FERMT1*. Kaplan–Meier survival curves were performed by the R package “survival” to analyze the relationship between *FERMT1* expression and the clinical survival prognosis of PAAD. In addition, the receiver operating characteristic (ROC) curve was performed using the R package “pROC” to assess the diagnostic value of *FERMT1*. Additionally, univariate and multivariate Cox regression analyses were applied to calculate the risk ratio of death for clinicopathological features and *FERMT1* expression to determine whether *FERMT1* could be used as an independent prognostic factor for PAAD. Cox regression analysis was conducted with the R package “survival”. The value of *p* < 0.05 was defined as statistically significant.

### 2.3. Methylation Analysis of FERMT1

The UALCAN database [21] (http://ualcan.path.uab.edu/, accessed on 15 September 2022) was performed to analyze the relationship between *FERMT1* and clinically relevant features, while the cBioPortal database (https://www.cbioportal.org/, accessed on 15 September 2022) verified DNA methylation of *FERMT1* in PAAD. Then, the MethSurv database [22] (https://biit.cs.ut.ee/methsurv/, accessed on 20 September 2022) was performed to evaluate DNA methylation levels and the prognostic value at the CpG sites of *FERMT1* in PAAD. Additionally, the correlation between *FERMT1* and 20 m^6^A-related genes [23] was also analyzed in PAAD samples from the TCGA-PAAD and GSE62452 datasets. Kaplan–Meier survival curves were conducted to analyze the prognostic value of m^6^A-related genes in PAAD samples. The data were presented using the R package “ggplot2”.

### 2.4. GO and KEGG Enrichment Analysis and Protein-Protein Interaction (PPI) Network Construction

The co-expression genes of *FERMT1* in the data of PAAD from TCGA were identified using the LinkedOmics database [24] (http://www.linkedomics.org/login.php, accessed on 25 September 2022) and Pearson correlation test. Then, the top 300 positively related genes were screened to perform Gene Ontology (GO) and Kyoto Encyclopedia of Genes and Genomes (KEGG) pathway enrichment analysis by Metascape [25] (http://metascape.org, accessed on 28 September 2022). The correlation index greater than 0.4, *p*-value less than 0.05, and false discovery rate (FDR) less than 0.05 were set up as thresholds. Moreover, the STRING database [26] (https://cn.string-db.org/, accessed on 28 September 2022) was utilized for the construction of the PPI network with a maximum confidence of 0.9 and visualized by Cytoscape [27] (http://www.cytoscape.org, accessed on 29 September 2022).

### 2.5. Gene Set Enrichment Analysis (GSEA)

The R package “DESeq2”and “clusterProfiler” were performed for GSEA analysis of the noteworthy functional and pathway of the differential expression genes (DEGs) between low and high expression *FERMT1* groups [28]. The c2.cp.KEGGv2022.1.Hs.symbols.gmt [KEGG Pathway Database] were retrieved from MsigDB [29] (https://www.gsea-msigdb.org/gsea/msigdb/index.jsp, accessed on 29 September 2022). The genome arrangement was analyzed repeatedly a thousand times, and FDR < 0.25, and *p*-value < 0.05 were defined as statistically significant.

### 2.6. Correlation between FERMT1 and Necroptosis in PAAD

The correlation between *FERMT1* expression and necroptosis-driving genes (*RIPK1*, *RIPK3* and *MLKL*) [30] in PAAD was analyzed by the R package “survival”. Meanwhile, the expression differences of necroptosis-driving genes between the tumor and normal groups in PAAD were also investigated by Pearson’s method. 

### 2.7. Associations with Immune Infiltration and Immune Checkpoints

The relationship between *FERMT1* expression and Immune score, Stromal score and ESTIMATE score was estimated using the ESTIMATE algorithm [31]. Then, the degree of immune infiltration, and the relative enrichment score of 24 immune cells in pancreatic cancer were calculated by a single sample gene set enrichment algorithm (ssGSEA) [32], which was accomplished by the R package “GSVA”. Spearman’s (TIMER database) and Pearson’s (GEPIA2 database [33], http://gepia2.cancer-pku.cn/index.html, accessed on 9 September 2022) method demonstrated the correlation coefficient of the tumor-infiltrating immune cell markers. Eight key immune checkpoint genes, including *CD274* (*PD-L1*), *CTLA4*, *HAVCR2*, *LAG3*, *PDCD1*, *PDCD1LG2*, *TIGIT*, and *SIGLEC15*, were also investigated to compare the different expression levels in normal and tumor groups and the correlation with *FERMT1*. 

### 2.8. Drug Sensitivity Analysis

The correlation between *FERMT1* expression and drug sensitivity was calculated by Pearson’s method with data from the CellMiner database [34] (http://discover.nci.nih.gov/cellminer/, accessed on 2 October 2022), which contains 792 FDA-approved or clinical trial drugs.

### 2.9. Statistical Analysis

A paired *t*-test or Wilcoxon rank sum test was used to compare the difference between two groups. Correlation analyses were performed using Spearman’s or Pearson’s correlation test. R software (version 4.2.1) was implemented for all statistical data analyses.

## 3. Results

### 3.1. FERMT1 Expression and Its Relationship with Clinicopathological Parameters in PAAD

We first explored the expression of FERMT1 in multiple cancers according to the TIMER database. Compared with normal tissues, the mRNA expression level of FERMT1 was considerably increased in 12 types of cancer including PAAD (Figure 1A and Appendix A). Meanwhile, we further verified that FERMT1 was upregulated in PAAD tissues relative to adjacent tissues through TCGA, GEPIA2 and two GEO datasets (GSE28735 and GSE62452) (Figure 1B–D, Appendix A). Additionally, immunohistochemical results obtained from the HPA database revealed that the protein expression level of FERMT1 was significantly elevated in PAAD tissues (Figure 1E). Then, we further examined the association between FERMT1 expression and clinicopathological characteristics in PAAD. The result exhibited that FERMT1 overexpression was significantly correlated with T stage, histologic grade, and primary therapy outcome (Figure 2). Altogether, the above findings indicated that the mRNA and protein levels of FERMT1 were highly expressed in PAAD tissues and related to clinicopathological parameters.

### 3.2. High FERMT1 Expression Is Correlated with Unfavorable Prognosis in PAAD Patients

To investigate whether FERMT1 expression correlated with prognosis in PAAD patients, the Kaplan–Meier curve was performed to calculate the overall survival (OS), disease-specific survival (DSS), and progression-free survival (PFS). The results showed that high FERMT1 expression was significantly related to worse OS (HR = 1.59, *p* = 0.03), DSS (HR = 1.62, *p* = 0.044), and PFS (HR = 1.85, *p* = 0.002) (Figure 3A–C). Besides, ROC curve analysis suggested that FERMT1 exhibited a favorable diagnostic value in PAAD patients (AUC = 0.970) (Figure 3D). To determine whether FERMT1 was an independent prognostic factor for the survival of PAAD patients, we performed a Cox regression analysis. According to univariate Cox regression analysis (Table 1), FERMT1 expression was remarkably associated with OS (HR = 1.588, *p* = 0.030). Multivariate Cox regression analysis suggested that FERMT1 expression was an independent factor affecting the prognosis in PAAD, as well as N stage (HR = 2.479, *p* = 0.029) and Histologic grade (HR = 2.468, *p* = 0.041). Furthermore, a nomogram according to Cox analysis results was constructed to predict the prognosis of PAAD patients. The C-index of the nomogram was 0.629, illustrating that the prediction accuracy of this model was favorable (Figure 3E). In addition, calibration curves also demonstrated that FERMT1 expression exhibited a good capacity to estimate 1-, 2-, and 3-year OS rates (Figure 3F). Taken together, FERMT1 expression has promising diagnostic and prognostic value for patients with PAAD, and can serve as an independent prognostic factor for PAAD patients.

### 3.3. Methylation Analysis of FERMT1

Spearman and Pearson correlation analysis showed that FERMT1 expression levels were negatively associated with the methylation level (Figure 4A). We identified 13 DNA methylation sites of FERMT1 (Figure 4B), of which cg04242132 (HR = 2.335, *p* = 0.00192) was significantly related to poor prognosis (Table 2). Simultaneously, the methylation level of FERMT1 gradually decreased with the development of PAAD, and the lowest methylation status was implied in stage 1 and grade 1 tumors (Figure 4C,D). The above findings implied that the DNA methylation status of FERMT1 was bound up with the emergence and progression of PAAD.

Subsequently, we investigated whether the expression of FERMT1 can be regulated by m^6^A modification and evaluated the association between 20 m^6^A-related genes and FERMT1 in PAAD. The results revealed that, in the TCGA-PAAD and GSE62452 datasets, the expression of FERMT1 was substantially positively connected with six m^6^A-related genes (IGF2BP2, IGF2BP3, YTHDF1, YTHDF2, VIRMA, RBM15), and negatively correlated with ALKBH5 (Figure 5A, *p* < 0.05). Then, the correlation between the expression of FERMT1 and seven m^6^A-related genes were drawn by the scatter plot (Figure 5B). Moreover, a prognostic heatmap suggested that the expression of IGF2BP2, IGF2BP3, and HNRNPC was strongly correlated with the poor prognosis of PAAD (Figure 5C). Furthermore, we screened out common genes with positive correlation and prognosis using a Venn diagram, including IGF2BP2 and IGF2BP3 (Figure 5D,E). These results indicated that FERMT1 may be intimately linked to the m^6^A modification in PAAD, and the regulation with IGF2BP2 and IGF2BP3 may promote the development and prognosis of PAAD.

### 3.4. Functional Enrichment Analysis of FERMT1 Co-Expression Genes in PAAD

According to the LinkedOmics database, 1320 positively correlated genes of FERMT1 were identified in PAAD with a correlation index greater than 0.4 (Figure 6A). The heat maps showed the top 50 positively and negatively associated genes. (Figure 6B,C). GO enrichment results suggested that the co-expression genes were mostly enriched in epithelial cell differentiation, mitotic cell cycle, cell-cell junction, cell adhesion molecule binding, and cadherin binding (Figure 6D). KEGG pathway analysis indicated enrichment in pathways in cancer, and PI3K-Akt signaling pathway (Figure 6E). Interestingly, FERMT1 may also play a role in necroptosis in PAAD (Figure 6E). In addition, GSEA results indicated that FERMT1 expression was positively correlated with DNA replication, base excision repair, and cell cycle (Figure 6F), and showed a negative association with the T cell receptor signaling pathway, cytokine receptor interaction, chemokine signaling pathway, and the JAK-STAT signaling pathway (Figure 6G).

### 3.5. PPI Network Analysis

The top 300 genes most positively correlated with FERMT1 were identified to construct a PPI network based on the STRING database (Figure 7A), and seven hub genes were identified, including CDK1, TOP2A, KIF11, CCNB2, ASPM, BUB1, DLGAP5 (Figure 7B). All of the above seven hub genes with high expression exhibited poor prognosis in PAAD (Figure 7C,D).

### 3.6. FERMT1 Expression May Be Associated with Necroptosis in PAAD

Recently, necroptosis has been regarded as a form of inflammatory programmed cell death. Necroptosis may cause an immunosuppressive tumor microenvironment, making it a potential cancer therapy approach [35]. Interestingly, in the KEGG analysis above, FERMT1 was found to be related to necroptosis. Therefore, we calculated the correlation between FERMT1 and the expression degree of main necroptosis-driving genes through the TIMER database. The results suggested that FERMT1 was significantly positively correlated with MLKL (R = 0.592, *p* < 0.001), RIPK1 (R = 0.242, *p* < 0.001) and RIPK3 (R = 0.596, *p* < 0.001) (Figure 8A–C). Similarly, in the high FERMT1 expression group, MLKL, RIPK1 and RIPK3 were significantly overexpressed (*p* < 0.001) (Figure 8D). These findings suggested that FERMT1 may be involved in tumorigenesis by regulating necroptosis.

### 3.7. Immune Infiltration Analysis of FERMT1 in PAAD

Since the immune cells in the tumor microenvironment are intimately connected with tumor progression and prognosis, the relationship between FERMT1 expression and the degree of immune infiltration was investigated in PAAD. The results indicated that StromalScore, ImmuneScore, and EstimateScore were considerably lower in the high FERMT1 expression group (Figure 9A). Then, the relationship between FERMT1 expression and the infiltration level of 24 immune cell subtypes was examined by ssGSEA. It was found that FERMT1 expression was adversely associated with the infiltration of pDC, TFH, cytotoxic cells, CD8^+^T cells, T cells, NK cells, mast cells, iDC, Tgd, and B cells, while positively correlated with Th2 cells (Figure 9B). The analysis results of the TIMER database showed that FERMT1 expression was inversely associated with immune cell infiltration, particularly CD8^+^T cells, Mast cells, and monocytes (Figure 9C). Moreover, on the basis of TIMER and GEPIA2 databases, there was a strong association between FERMT1 expression and several gene markers of immune cells, including CD8^+^ T cells, M1 macrophages, neutrophils, dendritic cells, T cells, Th1 and Th2 (Table 3). In addition, the expression of FERMT1 was substantially positively associated with eight classical immune checkpoint molecules (Figure 9D). Taken together, these findings indicated that FERMT1 is intimately correlated with immune cell infiltration and might have value in predicting immunotherapeutic benefits.

### 3.8. Correlation between FERMT1 and Drug Sensitivity

We explored the correlation between FERMT1 expression and drug sensitivity based on the data from the CellMiner database; the top 16 anticancer drugs were screened (Figure 10A). The sensitivity value of different drugs between the low and high FERMT1 expression groups was exhibited in Figure 10B. We observed that FERMT1 expression was negatively associated with arsenic trioxide, P-529 (Palomid 529, PI3K/Akt/mTOR inhibitor), motesanib (multi-targeted tyrosine kinase inhibitor) and carboplatin, while positively correlated with Palbociclib (CDK4/6 inhibitor), AM-5992 (AMG-925, dual inhibitor of CDK4 and FLT3) and TAE-226 (focal adhesion kinase selective inhibitor).

## 4. Discussion

*FERMT1* is essential to maintain cell-matrix adhesion as an integrin-interacting protein. The overexpression of *FERMT1* was found in multiple tumors such as colon cancer, gastric cancer, oral squamous cell carcinoma, and nasopharyngeal carcinoma, and has been associated with metastasis and poor prognosis [9,10,11,12]. FERMT1 was highly expressed in epithelial non-small-cell lung cancer (NSCLC), but not in neuroendocrine NSCLC. Cell migration in vitro and tumor growth in vivo are suppressed by the ectopic expression of FERMT1 in NSCLC cells [36]. In PAAD cell lines, the elevated *FERMT1* level was associated with promoting cell migration and invasion [13]. Microarray analysis of fresh tumor samples from human pancreatic cancer patients implanted into severely combined immunodeficient (SCID) mice revealed that *FERMT1* mRNA was increased in patient-derived xenograft (PDX) tumors compared to normal mice [37]. However, the research on the role of *FERMT1* in PAAD is still insufficient. In this study, based on the TCGA-PAAD and GEO datasets, we demonstrated that the mRNA and protein expression of *FERMT1* in PAAD tissues was significantly up-regulated. Furthermore, high *FERMT1* expression was associated with PAAD patients’ clinical stage, histological grade and primary therapy outcome. Meanwhile, survival analysis revealed that *FERMT1* overexpression was connected with unfavorable OS, DSS, and PFI in PAAD patients. Nomogram model results demonstrated the prediction accuracy of 1-, 2-, and 3-year survival rates by *FERMT1* expression. The ROC curve analysis showed that *FERMT1* had a high diagnostic value for PAAD. According to the above analysis, we speculated that *FERMT1* could be a potential biomarker for both diagnostic and prognostic purposes in PAAD.

DNA methylation, the most widely studied epigenetic modification, occurs in almost all cancers and is closely related to tumorigenesis and development [38]. In PAAD, we observed that the *FERMT1* expression was inversely correlated with DNA methylation, which was consistent with the characteristics of the oncogene. Moreover, methylation levels at six CpG loci located within *FERMT1* were related to prognosis. These results suggested that the methylation status of *FERMT1* might have potential prognostic value.

M^6^A methylation is the most abundant post-transcriptional modification in eukaryote mRNA [23]. Previous investigations have confirmed that m^6^A serves a critical role in the carcinogenesis and progression of PAAD. However, there has been no research on *FERMT1* and m^6^A. Here, we demonstrated that 16 key m^6^A-related genes were considerably expressed and elevated in the high *FERMT1* expression group. Furthermore, the expression of *FERMT1* was substantially positively connected with *IGF2BP2*, *IGF2BP3*, *YTHDF1*, *YTHDF2*, *VIRMA* and *RBM15*. In addition, only the high expression of *IGF2BP2* and *IGF2BP3* was linked to poor prognosis in PAAD. As the stabilizers of m^6^A methylation, *IGF2BP2* and *IGF2BP3* can enhance the stability of *B3GNT6* and spermine synthase (*SMS*) mRNA and protein expression, respectively, to promote the progression of pancreatic cancer [39,40]. In light of these findings, we proposed that the *FERMT1* may be modified by m^6^A to enhance mRNA stability, thereby accelerating the development of PAAD.

The biological function of *FERMT1* in PAAD was further investigated. GO and KEGG enrichment analysis suggested that *FERMT1* was mainly associated with epithelial cell differentiation, mitotic cell cycle, cell-cell junction, cell adhesion molecule binding, and cadherin binding, and the PI3K-Akt signaling pathway in PAAD. In addition, GSEA results indicated that significantly enriched pathways contained DNA replication, base excision repair, and cell cycle (Figure 5F). The above biological functions and pathways have been reported to play a significant role in the progression of PAAD [41,42,43,44]. Furthermore, the PPI networks showed that the top seven central genes were screened, including *CDK1*, *ASPM*, *TOP2A*, *KIF11*, *DLGAP5*, *BUB1* and *CCNB2*. Existing studies confirmed that *FERMT1* binds to *CDK*, which is a crucial regulatory factor of cell cycle progression [45]. *ASPM* promotes the aggressiveness of PAAD by maintaining Wnt-β-catenin signaling [46]. The Wnt signaling pathway is involved in cell cycle regulation, tumorigenesis, and T cytokines enhancement [47,48]. To sum up, these findings implied that *FERMT1* may participate in the progression of PAAD through signal pathways related to the cell cycle.

Necroptosis is a novel type of programmed cell death that is characterized by a combination of apoptosis and necrosis [49]. It plays multiple roles in tumor progression, patient outcome, immune regulation, and treatment [49]. It is worth noting that according to the results of KEGG, we found that *FERMT1* was related to necroptosis. Necroptosis is mainly regulated by *RIP1*, *RIP3*, and *MLKL*. The aurora kinase inhibitor, CCT137690, triggered the necroptosis of pancreatic cancer cells by *RIPK1*, *RIPK3*, and *MLKL*, and hindered the growth of in situ pancreatic tumors in mice [50]. In kidney tubular cells, RIPK3 and MLKL were found to be modulated by an integrin-linked kinase (ILK), a critical scaffold protein that contributed to necroptosis [14]. FERMT1 can interact with ILK and participate in integrin signal transduction, thus regulating a variety of biological functions such as adhesion, proliferation, and migration [15]. Subsequently, the correlation between *FERMT1* and these three core necroptosis-driving genes (*RIPK1*, *RIPK3*, and *MLKL*) was all significantly positive, which suggested that *FERMT1* might play a significant role in regulating the development of necroptosis through interacting with ILK in PAAD.

Infiltrating immune cells in the tumor microenvironment have a strong influence on the development and progression of pancreatic cancer [51]. In the results of GSEA, we observed that the *FERMT1* showed a negative correlation with the T cell receptor signaling pathway, cytokine receptor interaction, chemokine signaling pathway, and JAK-STAT signaling pathway (Figure 5G), which implied *FERMT1* may contribute to immune regulation. Our results demonstrated that the expression of *FERMT1* was positively connected with the infiltration level of Th2 cells, but negatively correlated with CD8^+^ T, NK cells, etc. High levels of CD4^+^/CD8^+^ T cells are significantly associated with favorable prognosis in PAAD patients, while Th2 cells are the opposite [52,53]. Simultaneously, Significant associations between *FERMT1* and these above immune cell gene markers were also revealed in PAAD. CD8^+^ T and NK cells are considered to be the main anti-tumor immune cells, which partly explained the worse prognosis of high *FERMT1* expression in PAAD patients. In addition, *FERMT1* expression was positively linked to immune checkpoint markers, such as *PD-L1* and *CTLA4*. In recent years, immune checkpoint inhibitors (ICIs) have achieved considerable success in a variety of cancers, including melanoma and lung cancer, though their clinical treatment efficacy in PAAD is limited [54]. It is worth pointing out that the combination of ICIs with radiotherapy and/or chemotherapy has shown encouraging results [55,56]. Palbociclib is a CDK4/6 inhibitor that can inhibit cell cycle progression and has been shown to regulate the immune characteristics of pancreatic cancer cells in patient-derived xenograft models [57]. In a survey of drug sensitivity in pancreatic cancer cell lines, E2F target scores were highly correlated with the AUC of Palbociclib sensitivity [58]. Preclinical evaluation of AMG 925 (i.e., AM-5922, a dual FLT3/CDK4 kinase inhibitor) in acute myeloid leukemia (AML) has demonstrated that the antitumor activity of AMG 925, was correlated with the inhibition of the pharmacodynamic markers STAT5 and RB phosphorylation, and also effective in suppressing FLT3 inhibitor-resistant mutants [59]. FERMT1 forms molecular complexes with focal adhesion kinase (FAK), β1-integrins, α-actinin and migfilin to modulate cell shape and migration, while TAE226, a FAK inhibitor, has inhibitory effects on the growth of a variety of tumor cells. [60,61] However, whether FERMT1 functions either as a target of the above-mentioned drugs in PAAD or in association with drug resistance has not been reported. Here, our results indicated that patients with high *FERMT1* expression were more sensitive to Palbociclib, AM-5992 and TAE-226, which could serve as drug candidates for combination therapy to improve pancreatic cancer treatment.

As far as we know, this is the first systematic exploration of the connection between the expression of *FERMT1* and m^6^A, necroptosis, and immune infiltration in PAAD. However, this study still has some limitations. Firstly, the expression data from several public databases may have systematic errors. Secondly, the potential biological function and regulatory mechanism of *FERMT1* in PAAD requires further experimental validation. Finally, there were no complete cases of the effectiveness of candidate drugs against *FERMT1*. In future research, we will perform more experiments to verify these results.

## 5. Conclusions

In summary, this study reveals that *FERMT1* is highly expressed in PAAD and associated with clinicopathological features. Elevated expression of *FERMT1* is correlated with an unfavorable prognosis in PAAD patients. The DNA methylation of *FERMT1* is connected with the prognosis of PAAD, and *FERMT1* expression is closely related to m^6^A modification and the regulation of necroptosis. Besides, high expression of *FERMT1* is not only associated with immune infiltration and immune checkpoint genes but also sensitive to some chemotherapy drugs, which means that *FERMT1* may be suitable as a potential target for combination therapy. *FERMT1* is a promising biomarker for PAAD diagnosis and prognosis evaluation theoretically, however, more clinical samples and functional mechanism studies are needed to verify the application of FERMT1 in diagnosis, prognosis evaluation and target therapy of PAAD, so as to accelerate its promotion in future clinical trials.

## Figures and Tables

**Figure 1 genes-14-00734-f001:**
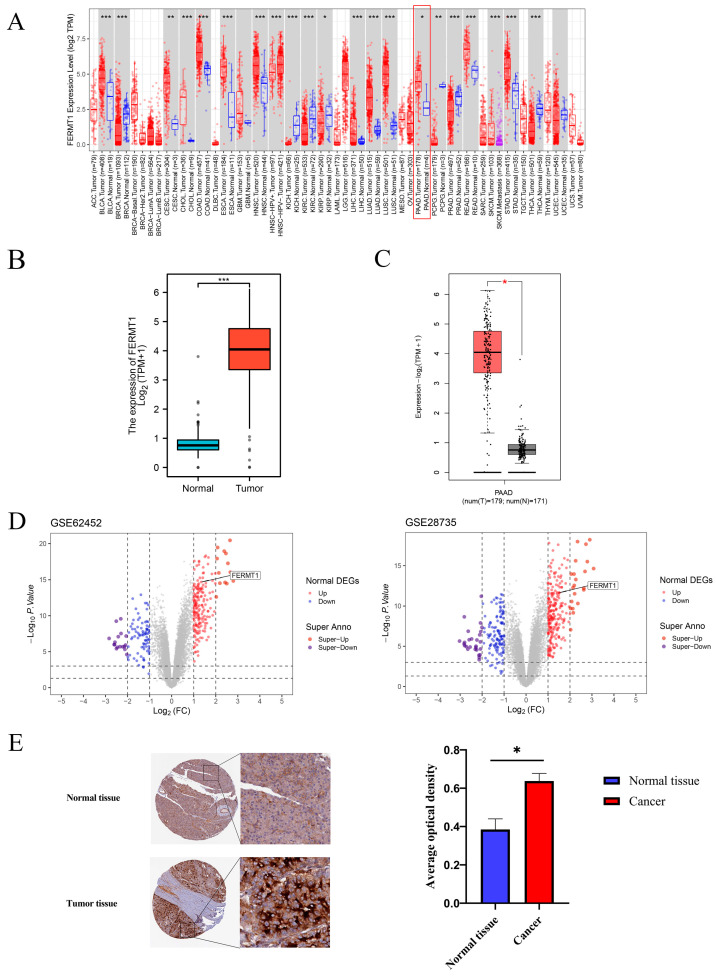
The expression of FERMT1 in pan-cancer and PAAD. (**A**) FEMRT1 expression in different cancers based on TIMER database. (**B**) FERMT1 expression levels in PAAD and corresponding normal tissues in TCGA-PAAD, (**C**) GEPIA 2 and (**D**) GEO datasets (GSE62452 and GSE28735). (**E**) FERMT1 protein level in PAAD and adjacent normal tissues determined according to the HPA database. * *p* < 0.05, ** *p* < 0.01, *** *p* < 0.001.

**Figure 2 genes-14-00734-f002:**
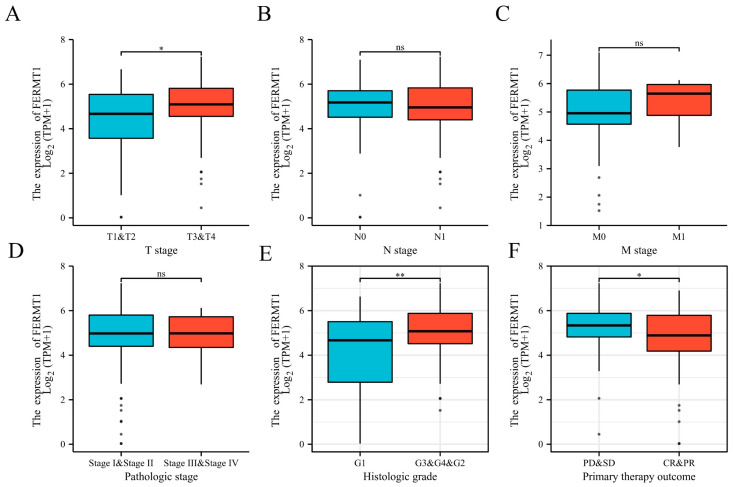
The correlation between FERMT1 expression and clinicopathological parameters. High FERMT1 expression was significantly related to (**A**) T stage, (**B**) N stage, (**C**) M stage, (**D**) pathological stage, (**E**) histological type, (**F**) primary therapy outcome. PD, progressive disease; SD, stable disease; CR, complete response; PR, partial response. * *p* < 0.05, ** *p* < 0.01, ns, no significant difference.

**Figure 3 genes-14-00734-f003:**
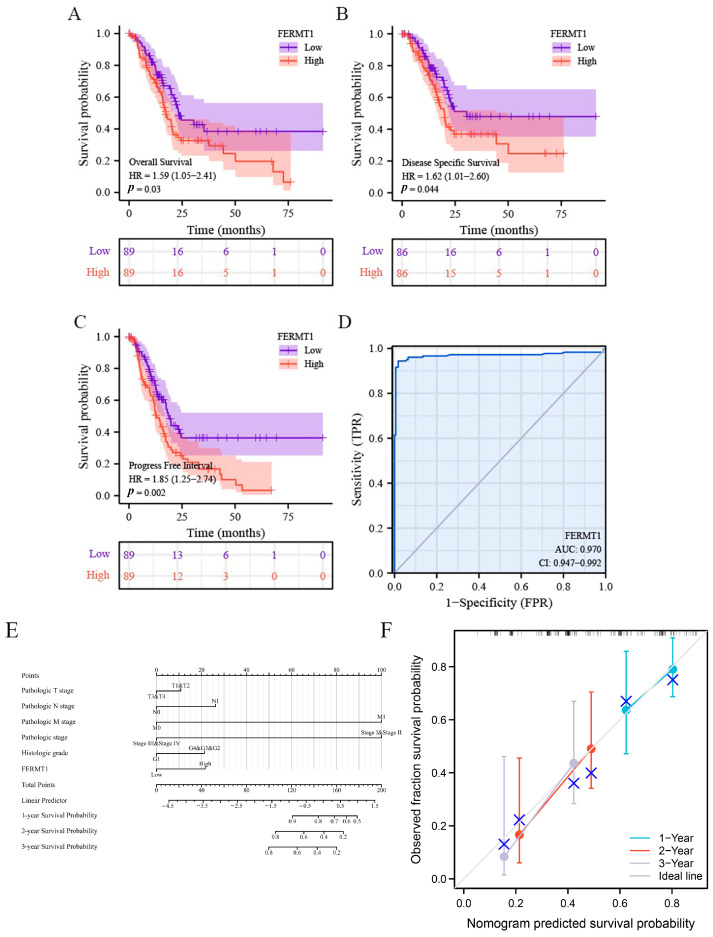
The prognostic value of FERMT1 expression in PAAD patients. (**A**) OS, (**B**) DSS and (**C**) PFI analysis in the Kaplan–Meier Plotter database based on TCGA data. (**D**) The ROC curve was performed to assess the diagnostic efficiency of FERMT1 in PAAD. (**E**) A nomogram and (**F**) calibration curves for estimating 1-, 2-, and 3-year OS of PAAD patients.

**Figure 4 genes-14-00734-f004:**
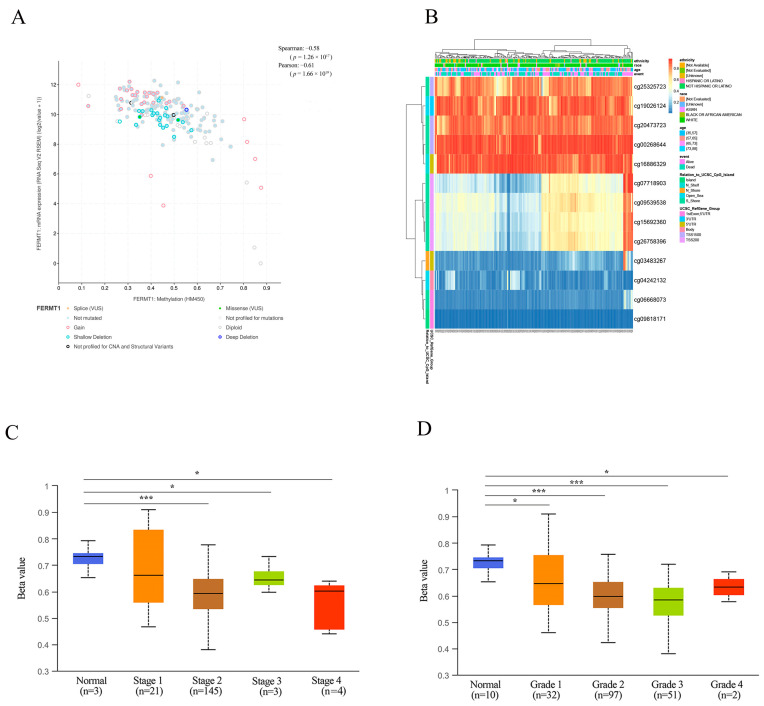
Correlation of FERMT1 with DNA methylation in PAAD. (**A**) Spearman and Pearson correlation analysis between FERMT1 expression and DNA methylation. (**B**) The heatmap of DNA methylation at CpG sites of FERMT1. (**C**, **D**) The promoter methylation level of FERMT1 based on tumor stage (**C**) and grade (**D**). * *p* < 0.05, *** *p* < 0.001.

**Figure 5 genes-14-00734-f005:**
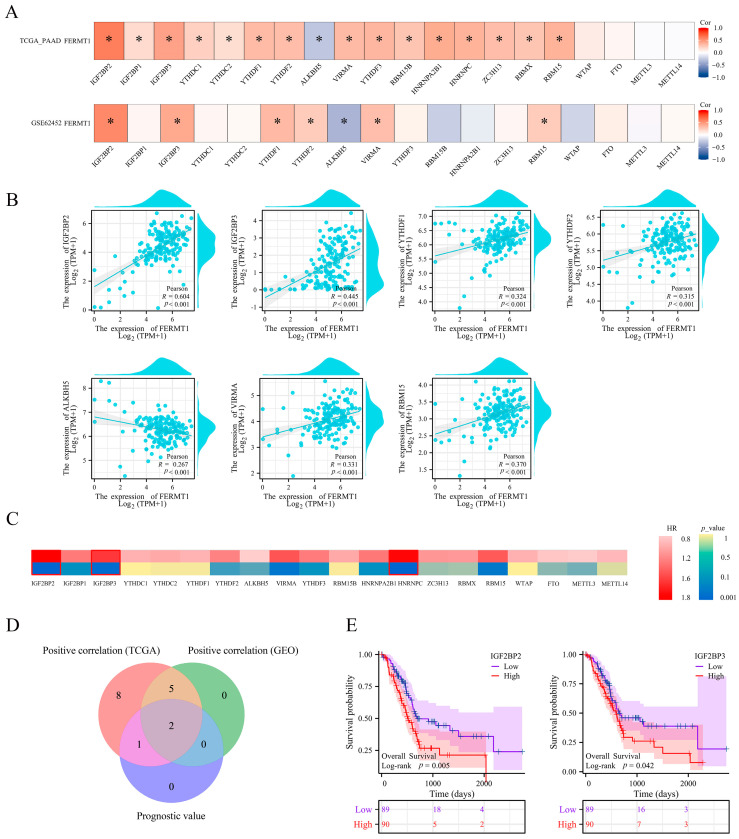
Correlation analysis of *FERMT1* expression with m^6^A-related genes in PAAD. (**A**) Correlation between *FERMT1* and m^6^A-related gene expression in PAAD was analyzed using the TCGA-PAAD and GSE62452 datasets. (**B**) The correlation between *FERMT1* and the expression of m^6^A-related genes were described using scatter plots, including *IGF2BP2*, *IGF2BP3*, *YTHDF1*, *YTHDF2*, *ALKBH5*, *VIRMA* and *RBM15*. (**C**) Prognostic heatmap of m^6^A-related genes. (**D**) Venn diagram showed common genes with positive correlation and prognosis. (**E**) Kaplan-Meier curves of *IGF2BP2* and *IGF2BP3*. * *p* < 0.05.

**Figure 6 genes-14-00734-f006:**
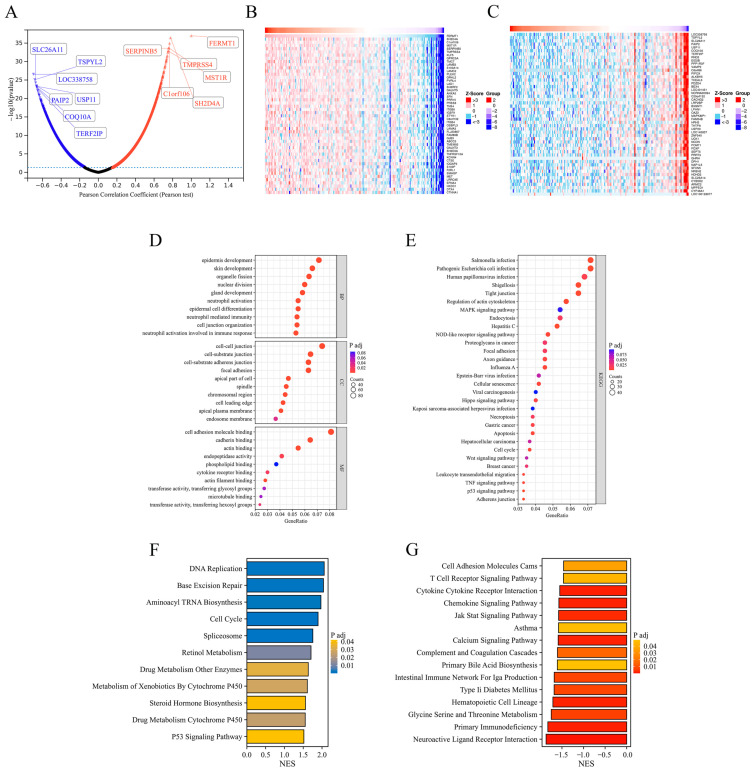
Functional enrichment analysis of FERMT1 co-expression genes in PAAD. (**A**) The volcano map of genes differentially expressed in correlation with FERMT1. (**B**) The top 50 positive co-expression genes. (**C**) The top 50 negatively co-expression genes. (**D**) The GO analysis of co-expression genes in molecular function (MF), cellular component (CC), and biological process (BP). (**E**) KEGG pathway analysis of co-expression genes. (**F**) GSEA enrichment analysis in high FERMT1 expression group. (**G**) GSEA enrichment analysis in low FERMT1 expression group.

**Figure 7 genes-14-00734-f007:**
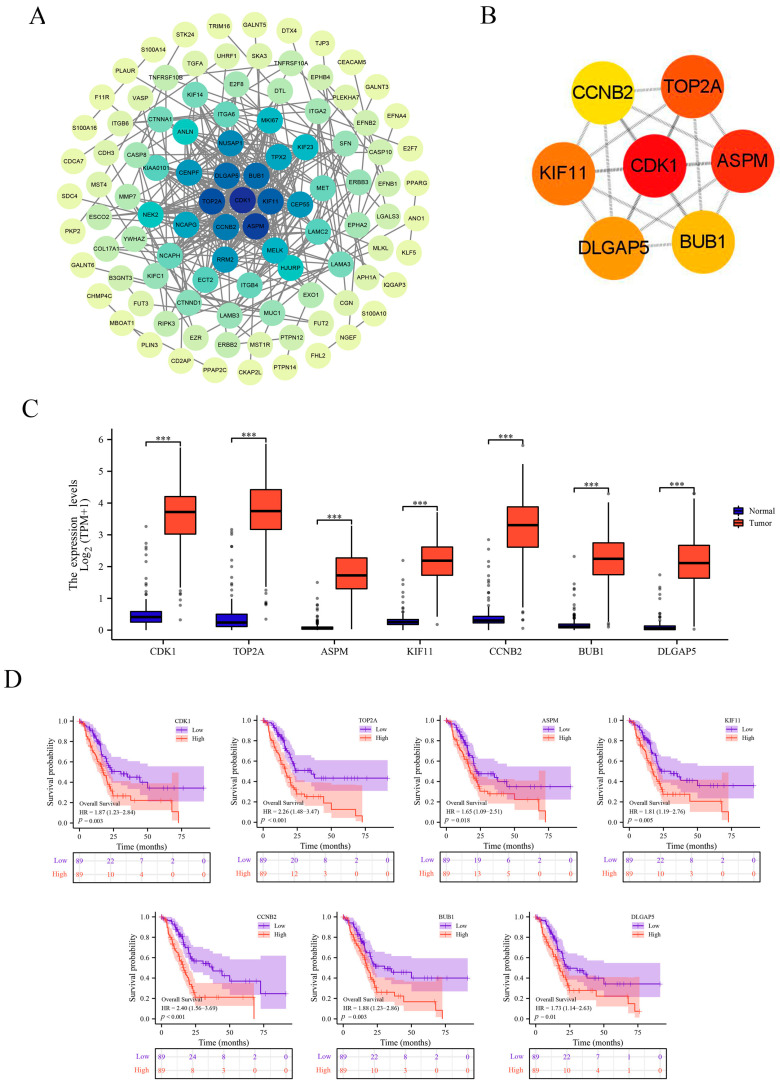
PPI network and hub genes analysis. (**A**) PPI network of top 300 positively related genes. (**B**) hub genes. (**C**) The mRNA expression of hub genes in PAAD. (**D**) The prognostic analysis of hub genes in PAAD. *** *p* < 0.001.

**Figure 8 genes-14-00734-f008:**
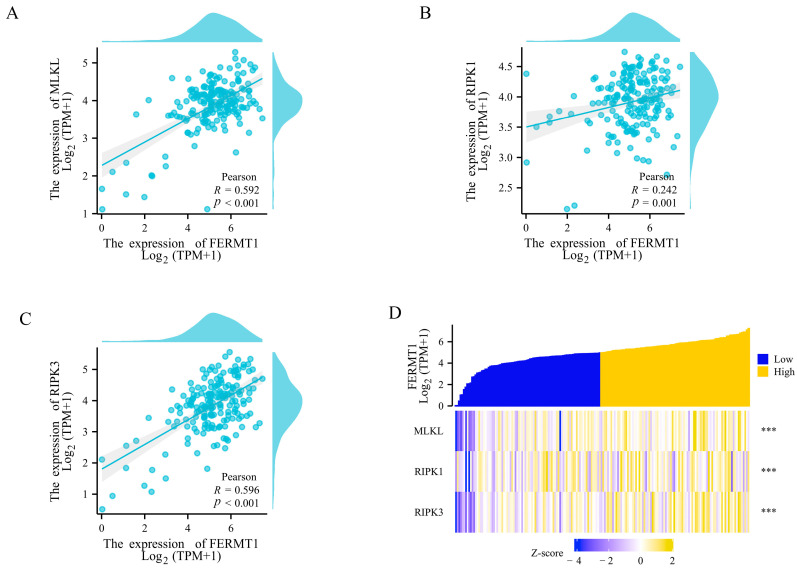
The correlation between FERMT1 expression and necroptosis-driving genes in PAAD. (**A**) MLKL, (**B**) RIPK1, (**C**) RIPK3, (**D**) Expression differences of necroptosis-driving genes in the low and high FERMT1 expression groups. *** *p* < 0.001.

**Figure 9 genes-14-00734-f009:**
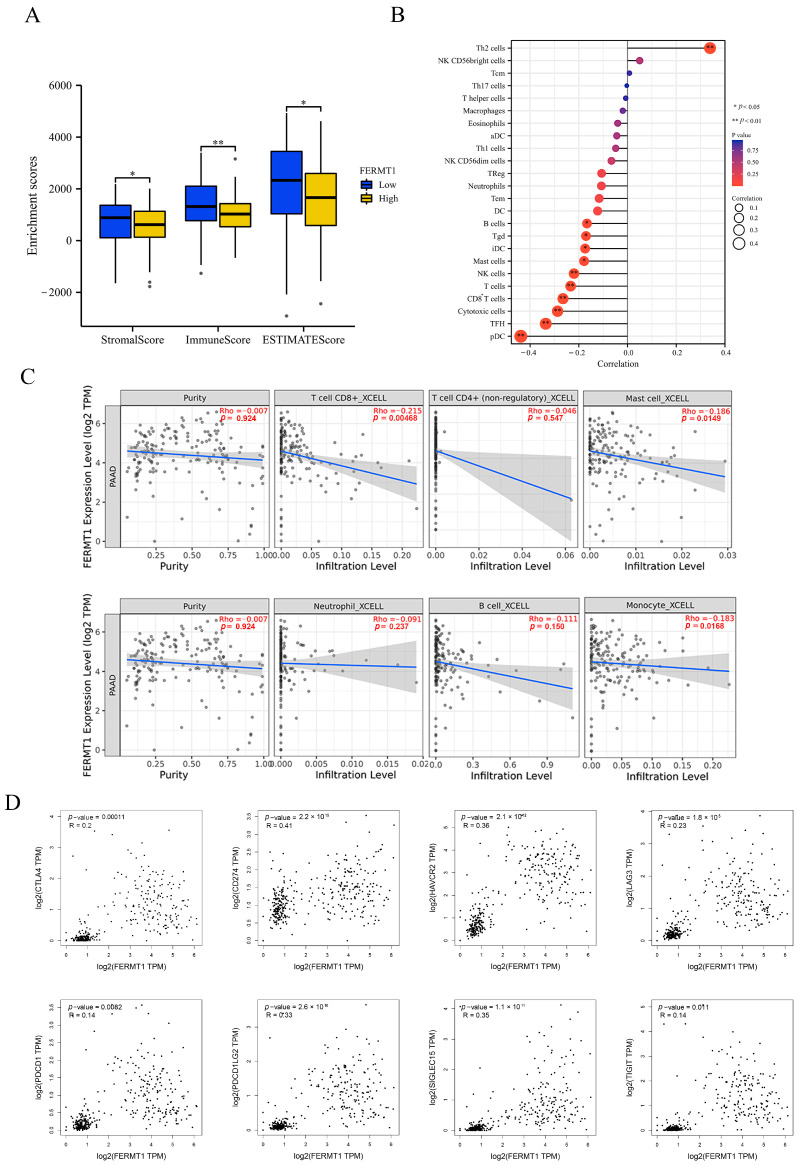
The relationship between FERMT1 expression and immune infiltration in PAAD. (**A**) StromalScore, ImmuneScore, and ESTIMATEScore analysis in high/low FERMT1 expression group. (**B**) Correlation between FERMT1 expression and 24 immune cell subtypes by ssGSEA. (**C**) Association between FERMT1 expression and immune cell infiltration by TIMER database. (**D**) Scatter plots showing the correlation between FERMT1 expression and eight immune checkpoint-related genes. * *p* < 0.005; ** *p* < 0.01.

**Figure 10 genes-14-00734-f010:**
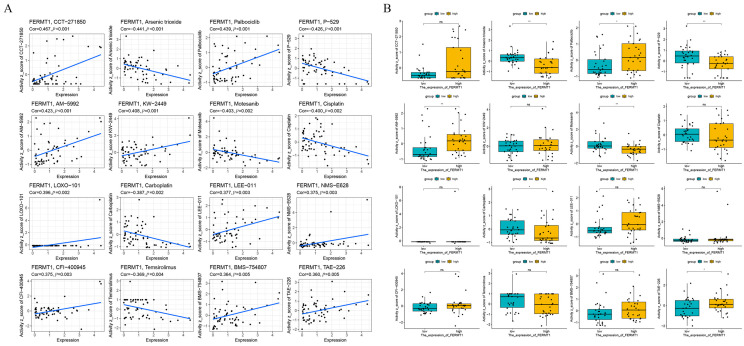
Correlation between FERMT1 and drug sensitivity. (**A**) Correlation between FERMT1 and the top 16 anti-cancer drugs with sensitivity in the CellMiner database. (**B**) The sensitivity value of different drugs between the low and high FERMT1 expression groups. * *p* < 0.05, ** *p* < 0.01, ns, no significant difference.

**Table 1 genes-14-00734-t001:** Univariate and multivariate Cox regression analysis of FERMT1 expression for overall survival in patients with PAAD.

	Univariate Analysis			Multivariate Analysis		
Characteristics	HR	95% CI	*p* Value	HR	95% CI	*p* Value
Age (≤65 vs. >65)	1.290	0.854–1.948	0.227			
Gender (Female vs. Male)	0.809	0.537–1.219	0.311			
T stage (T1&T2 vs. T3&T4)	2.023	1.072–3.816	0.030	1.597	0.571–4.463	0.372
M stage (M0 vs. M1)	0.756	0.181–3.157	0.701			
N stage (N1 vs. N0)	2.154	1.282–3.618	0.004	2.479	1.097–5.601	0.029
Pathologic stage (Stage I vs. Stage II)	2.325	1.065–5.073	0.034	0.458	0.101–2.058	0.312
Histologic grade (G1 vs. G2&G3&G4)	2.164	1.139–4.110	0.018	2.486	1.040–5.945	0.041
Residual tumor (R0 vs. R1&R2)	1.645	1.056–2.561	0.028	1.510	0.886–2.572	0.129
Primary therapy outcome (PD&SD vs. PR&CR)	0.425	0.267–0.677	<0.001	0.549	0.328–0.917	0.022
FERMT1 (Low vs. High)	1.588	1.046–2.410	0.030	1.326	0.836–2.101	0.030

**Table 2 genes-14-00734-t002:** The prognostic value of CpG in FERMT1.

Name	HR	CI	*p* Value	LR_Test_Pvalue	Best_Split	UCSC_RefGene_Group	Relation_to_UCSC_CpG_Island
cg03483267	0.521	(0.345; 0.786)	0.00187	0.001622	median	5′UTR	N_Shore
cg04242132	2.335	(1.367; 3.991)	0.00192	0.000771	q25	Body	Open_Sea
cg09539538	0.455	(0.264; 0.783)	0.004455	0.002044	q75	TSS200	S_Shore
cg15692360	0.477	(0.282; 0.808)	0.005878	0.003012	q75	TSS200	S_Shore
cg26758396	0.495	(0.288; 0.849)	0.01057	0.005922	q75	TSS200	S_Shore
cg07718903	0.554	(0.334; 0.916)	0.021503	0.015061	q75	TSS200	S_Shore

**Table 3 genes-14-00734-t003:** Correlation analysis between FERMT1 and immune cell markers using TIMER and GEPIA2 databases.

Immune Cells	Marker Genes	GEPIA-2 (Pearson)		TIMER (Spearman)	
		R	*p*	R	*p*
CD8^+^ T cells	CD8A	−0.27	0.00018	−0.214	0.00404
	CD8B	−0.28	0.00016	−0.205	0.00601
M1 macrophages	PTGS2	0.18	0.014	0.32	0.0000138
Neutrophils	CCR7	−0.23	0.0022	−0.205	0.00611
Dendritic cells	HLA-DPB1	−0.25	0.00071	−0.193	0.00963
T cell	CD3D	−0.26	0.00047	−0.19	0.0109
	CD3E	−0.22	0.0033	−0.195	0.00915
	CD2	−0.22	0.0028	−0.196	0.00869
Th1	TBX21	−0.3	0.000047	−0.269	0.000281
	STAT4	−0.37	0.0000002	−0.372	0.00000035
	STAT1	0.20	0.0057	0.215	0.00391
	IL12A	−0.19	0.01	−0.181	0.0154
Th2	STAT6	0.3	0.000041	0.357	0.00000112
	IL13	−0.16	0.026	−0.243	0.00104

## Data Availability

Not applicable.

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
