# Peer review of "FERMT1 Is a Prognostic Marker Involved in Immune Infiltration of Pancreatic Adenocarcinoma Correlating with m6A Modification and Necroptosis"

_genes, 2023, doi:10.3390/genes14030734_

Round 1

Reviewer 1 Report

Pancreatic adenocarcinoma (PAAD) is a particularly aggressive form of cancer with a very poor prognosis. Despite advances in surgical techniques and chemotherapy, the survival rate for pancreatic cancer remains low. This highlights the need for further research in order to understand the biology of pancreatic cancer and develop more effective treatments for this disease.

Wu and colleagues focused in this work on the role of FERMT1, a protein that, according to previous studies, plays a key role in cell invasion, migration and metastasis in different cancer entities, including PAAD; however, no diagnostic/prognostic value could be attributed so far to this marker in this context. With the present study, the author proposed a comprehensive in silico approach describing the clinical role of FERMT1 in PAAD patients exploiting multiple public databases.

The reported analyses described various molecular (transcriptome, epigenome, post-translational modifications, interactome), functional and microenvironmental characteristics linked to the role of FERMT1 in PAAD.

Overall, the manuscript reads well, presenting a solid and linear structure; results, conclusions and limitations are clearly and adequately described.

Prior to acceptance, the manuscript could be improved by addressing the following minor comments:

-The relationship between FERMT1 and m6A modifications should be better described. It is not immediately clear based on which previous insights the authors hypothesized that FERMT1 expression could be regulated by this specific chemical modification. Moreover, the conclusion that FERMT1 directly undergoes m6A modifications since many m6A-related genes are upregulated in high FERMT1 expression PAADs sounds weak;

-FERMT1 and necroptosis: is it possible to describe an underlying mechanism that links FERMT1 expression/activity to the MLKL/RIPK1/RIPK3-mediated necroptosis pathway? This would greatly help to understand the impact of FERMT1 expression impact in this mechanism;

-Figure 3A: Are here the “FERMT1” methylation levels plotted in the graph corresponding to the average of the beta values of the various probes overlapping FERMT1 or was it considered only the intensity value of one specific probe per each case?

-Figure 3A and 3B should be replaced by high-quality panels as Fig3A is clearly cropped and the legend of Fig3B is very difficult to read;

-Figures 5 B and C: What are the “Group” legends referring to? These plots appear difficult to interpret;

- Figure 1D: Here, a distinction between “normal DEGs” and “Super Anno” has been proposed, although it was not provided an explanation of those two terms. It would help to clarify these annotations either in the figure legend or in the material and methods section;

Figure 2F: The text reported on the x-axis (“PD&SD”; “CR&PR”) is not properly described in any part of the text;

-Table 3: Is there a specific reason why two different statistical measures have been adopted for the two databases? What is the meaning of the asterisk on the Pearson value for IL13 in GEPIA-2 database (0.026)?

- Text needs a little editing for better clarity since some typos or punctuation issues could be detected.

Reviewer 2 Report

Wu et al. address the relevance of FERMT1 as a diagnostic and Prognostic Marker of pancreatic Adenocarcinoma in the context of Immune Infiltration and modification of nucleosides such as m6A. 

However, the importance of hallmarks such as modification of nucleosides, and necroptosis are meaning for in the cancer initiation and progression.

But, this paper needs a better presentation and a focussed view. 

Important suggestions that can help for better impact.

1. In the introduction, there is no link for the basis of FERMT1 association with PAAD, modified nucleiosides, and necroptosis.

2. Discussion of FERMT1 and metabolic adaptations during chemotherapy drug responses may be encouraged.

3. Some data need to be generated on the expression of FERMT1 in PAAD.

4. A discussion of is important heterogeneity of FERMT1 across PAAD to other tumor types.

5. Preclinical and clinical scope may be highlighted.

Round 2

Reviewer 2 Report

The authors have addressed majority of suggestions.